# Optimization of Naringenin Nanoparticles to Improve the Antitussive Effects on Post-Infectious Cough

**DOI:** 10.3390/molecules27123736

**Published:** 2022-06-10

**Authors:** Zhengqi Dong, Xiangtao Wang, Mingyue Wang, Rui Wang, Zheng Meng, Xiaotong Wang, Bo Yu, Meihua Han, Yifei Guo

**Affiliations:** 1Institute of Medicinal Plant Development, Chinese Academy of Medical Sciences & Peking Union Medical College, No. 151, Malianwa North Road, Haidian District, Beijing 100193, China; zqdong@implad.ac.cn (Z.D.); xtaowang@163.com (X.W.); wmy313714384@163.com (M.W.); mz960428@163.com (Z.M.); ttwang0521@163.com (X.W.); yywyzb@163.com (B.Y.); hanmeihua727@163.com (M.H.); 2Key Laboratory of Bioactive Substances and Resources Utilization of Chinese Herbal Medicine, Ministry of Education, Chinese Academy of Medical Sciences & Peking Union Medical College, Beijing 100094, China; 3Beijing Key Laboratory of Innovative Drug Discovery of Traditional Chinese Medicine (Natural Medicine) and Translational Medicine, Chinese Academy of Medical Sciences & Peking Union Medical College, No. 151, Malianwa North Road, Haidian District, Beijing 100193, China; 4College of Pharmacy, Heilongjiang University of Chinese Medicine, Harbin 150040, China; wrdx@sina.com

**Keywords:** naringenin, nanoparticle, antitussive effect, antioxidant effect, anti-inflammatory effect

## Abstract

Naringenin (NRG) is a natural compound with several biological activities; however, its bioavailability is limited owing to poor aqueous solubility. In this study, NRG nanoparticles (NPs) were prepared using the wet media milling method. To obtain NRG NPs with a small particle size and high drug-loading content, the preparation conditions, including stirring time, temperature, stirring speed, and milling media amount, were optimized. The NRG (30 mg) and D-α-tocopherol polyethylene glycol succinate (10 mg) were wet-milled in deionized water (2 mL) with 10 g of zirconia beads via stirring at 50 °C for 2 h at a stirring speed of 300 rpm. As a result, the NRG NPs, with sheet-like morphology and a diameter of approximately 182.2 nm, were successfully prepared. The NRG NPs were stable in the gastrointestinal system and were released effectively after entering the blood circulation. In vivo experiments indicated that the NRG NPs have good antitussive effects. The cough inhibition rate after the administration of the NRG NPs was 66.7%, cough frequency was three times lower, and the potential period was 1.8 times longer than that in the blank model group. In addition, the enzyme biomarkers and histological analysis results revealed that the NRG NPs can effectively regulate the inflammatory and oxidative stress response. In conclusion, the NRG NPs exhibited good oral bioavailability and promoted antitussive and anti-inflammatory effects.

## 1. Introduction

Naringenin (NRG) is a flavonoid from the flavanone family mostly found in citrus fruits and tomato skin [1,2], and presents several bioactivities, including antitussive [3,4], anti-inflammation [5], antioxidant [6], antibacterial [7], cardiovascular protection [8], and antitumor effects [9]. However, as a hydrophobic bioactive agent, it has poor aqueous solubility resulting in low bioavailability [10,11], which severely limits its therapeutic efficacy [12,13]. Thus, to improve its bioavailability, nanotechnology has been used to construct the NRG nanodelivery systems [2], including nanosuspensions [14], vesicles [15], nanoparticles [16], nanomicelles [17], nanoemulsions [18], nanocomplexes [19], and nanocomposites [20].

Many biosafe and biodegradable materials have been utilized as nanocarriers to prepare the nanodelivery system for NRG, in particular, chitosan [21,22], cyclodextrin [23,24], polylactic acid [25,26], polylactic acid/glycolic acid copolymer [27,28], polycaprolactone [29,30], poly(vinylpyrrolidone) [31], and tocopheryl polyethylene glycol succinate (TPGS) [32,33]. Among them, TPGS has been utilized broadly as a nanocarrier to encapsulate NRG, because of its excellent amphiphilic properties, entrapment efficacy, and biosafety [34,35]. In our previous study, we prepared the NRG nanoparticles (NPs) with TPGS as a drug carrier via wet media milling, and the obtained NRG NPs exhibited good antitussive and expectorant effects [36]. 

It is well known that several factors, such as the temperature, the size and amount of milling beads, milling time, and milling speed, affect the physicochemical properties of the nanoparticles during wet media milling [37,38]. Moreover, the critical formulation parameter (the ratio of the drug to the nanocarriers) was reported to significantly affect the physicochemical properties of nanoparticles [39,40]. Therefore, the process parameters should be optimized on a case-by-case basis to obtain the NRG NPs with excellent therapeutic efficacy.

In this study, the NRG NPs were prepared using TPGS as the nanocarrier via the wet media milling method. The preparation procedure was optimized to obtain the NRG NPs with a high drug-loading content and small particle size. The ratio of the NRG to the TPGS, the temperature, time, speed, and the amount of the milling beads were studied. After optimizing the preparation parameters, the physicochemical properties of the NRG NPs were evaluated. A cough is a clinical symptom of inflammation in infectious diseases [41]; therefore, the antitussive effect of the NRG NPs on post-infectious cough was studied, and several relative enzyme biomarkers were measured to prove the antitussive effect.

## 2. Results and Discussion

### 2.1. Optimized Preparation Method

According to previous reports, the drug-loading content (DLC) is significantly affected by the feed–weight ratio (FWR) of the drug to the carrier [42,43]. Hence, to obtain the optimal DLC, the drug/carrier ratios of 1/1, 2/1, 3/1, 4/1, 6/1, and 8/1 were conducted in this study, and the results are shown in Table 1. When the FWR of the drug to carrier was below 4/1, the naringenin (NRG) nanoparticles (NPs) were successfully prepared; when the FWR reached 6/1, the NRG NPs were not obtained, and a large amount of precipitation was observed. This was attributed to the encapsulation efficiency; when the drug amount exceeded the encapsulation efficiency of the tocopherol polyethylene glycol succinate (TPGS) [44], the hydrophobic drug could not be completely entrapped in the nanoparticles, and would aggregate to precipitation owing to the low aqueous solubility. The increase in the FWR of the NRG/TPGS from 1/1 to 4/1 significantly affected the DLC. In particular, with the increase in the FWR from 1/1 to 3/1, the DLC promoted from 42.3% to 66.1% (60.0% vs. 42.3%, *** *p* < 0.001; 66.1% vs. 60.0%, ^##^
*p* < 0.01). A further increase in the FWR to 4/1, caused a slight decrease in the DLC (64.5%), which was not statistically different (64.5% vs. 66.1%, *p* > 0.05) and was also induced by the encapsulation efficiency of the TPGS [45]. In addition, the average particle size of the NRG NPs was approximately 220 nm. Considering the DLC and the particle diameter, the drug/carrier ratio of 3/1 was selected as the optimum for the preparation of the NRG NPs.

The preparation conditions were reported to affect the physicochemical properties of the nanoparticles; hence, several factors, including stirring time, temperature, stirring speed, and the amount of media, were estimated [37]. The particle size and the polydispersity index (PDI) were utilized as the evaluation criteria [46,47,48]; all of the results are shown in Table 2. To study the effect of the stirring time, the samples were stirred for 1, 2, 3, and 4 h. The particle size decreased and then increased with the increasing stirring time (Table 2, samples 1–4), and the best particle size and PDI were obtained when the samples were stirred for 2 h (Table 2, sample 2). Then, the effect of the temperature was evaluated; the samples were prepared at 0, 25, 50, and 60 °C (Table 2, samples 5–8). The results indicated that the temperature affected the particle size of the NRG NPs; particularly, at 50 °C, the smallest particle diameter was obtained (Table 2, sample 7). Next, the effect of the stirring speed was researched; the particle size of the NRG NPs significantly decreased with the increasing stirring speed, from 100 to 400 rpm (Table 2, samples 9–12), and the NRG NPs with the best particle diameter were obtained at a stirring speed of 300 rpm (Table 2, sample 11). Finally, the number of zirconia beads was studied (Table 2, samples 13–16), and the results indicated that the NRG NPs had a better particle diameter when the 10 g zirconia beads were utilized (Table 2, sample 15). Based on these results, to obtain the NRG NPs with the smallest particle size and suitable PDI, the NRG (30 mg) and TPGS (10 mg) were dispersed in deionized water (2 mL), transferred to a vial containing 10 g zirconia beads, and stirred at 50 °C for 2 h at a stirring speed of 300 rpm.

### 2.2. Particle Size and Morphology

After optimizing the preparation conditions, the NRG NPs with a hydrodynamic diameter of 182.2 nm (PDI ~0.28) were obtained (Figure 1a). Transmission electron microscopy (TEM) results showed that the NRG NPs presented a sheet-like morphology (Figure 1b), which is consistent with our previous report [36].

### 2.3. Crystallographic Analysis

Drugs have been reported to lose their crystallinity when entrapped by nanocarriers. To verify the form of the NRG, the NRG NPs were detected by differential scanning calorimetry (DSC). The NRG powder, TPGS, and their physical mixtures were measured under the same conditions (Figure 2a). The NRG powder and physical mixture presented an endothermic peak at 250 °C, which is the melting point of the NRG. This endothermic peak was not observed in the NRG NPs, which was consistent with previously reported results [49], suggesting that there were molecular interactions between the NRG and TPGS. The crystal forms of these samples were estimated using X-ray diffraction (XRD, Figure 2b). The NRG powder and the physical mixture presented significant intense NRG peaks, suggesting its crystallinity. In contrast, the characteristic diffraction peaks of the NRG disappeared in the NRG NPs, indicating that the NRG was amorphous or molecularly dispersed in the NRG NPs [50]. The DSC and XRD results verified that the crystalline form of the NRG changed during the preparation process, owing to the interaction between the NRG and TPGS.

### 2.4. Study on Drug Release In Vitro

The release properties of the NRG NPs were studied in vitro using a sequence of release media that would imitate release in vivo: artificial gastric juice; artificial intestinal fluid; and phosphate-buffered saline (PBS) (Figure 3). Within the initial 4 h, only 8.4% of the NRG was released from the NRG NPs in artificial gastric juice. Then, the release medium was changed to artificial intestinal fluid, and in the following 8 h, the release rate was also slow, only 16.9% of the NRG was released. The release medium was then changed to PBS; in the following 88 h, the release rate was enhanced significantly, and approximately 95.6% of the NRG was released. In contrast, the NRG powder exhibited a low release rate, with approximately 40% of NRG released during the entire procedure. These results suggest that the NRG NPs can be stable in the gastrointestinal system (a small amount of the NRG was released during the procedure), whereas, after entering the blood circulation, the NRG could be released effectively. According to these results, the aqueous solubility and bioavailability of the NRG can be significantly promoted via encapsulation in NPs.

### 2.5. Antitussive Effect

As a well-known natural compound with a wide spectrum of biological activities, NRG was previously utilized to treat coughs [1,2]. In the present study, the antitussive effects of the NRG NPs were evaluated using a post-infectious cough (PIC) model. The mice were divided into four groups: blank model (normal saline); positive (montelukast sodium); the NRG; and the NRG NPs. The cough frequency in 5 min and the cough incubation period of all groups were recorded 24 h after the last intragastric administration to estimate the antitussive effect of the NRG NPs. After 5 min, the cough frequencies of 36 ± 3, 15 ± 3, 19 ± 1, and 12 ± 3 times were recorded for the blank model, the positive drug group, the NRG, and the NRG NPs groups, respectively (Figure 4a). Compared to the blank model group, the positive drug presented a good antitussive effect, with a cough inhibition rate of 58.3%. The NRG group showed a slightly lower antitussive rate than the positive drug group (47.2%), but no significant difference was shown. The NRG NPs exhibited excellent antitussive effect; the cough frequency decreased three-fold compared with the blank model (12 vs. 36, *** *p* < 0.001) with an inhibition rate of 66.7%. Furthermore, the NRG NPs showed a better antitussive effect than the NRG; the cough frequency was decreased 1.6-fold (12 vs. 19, ^#^
*p* < 0.05), and, compared with the positive drug, the NRG presented a similar antitussive effect to the positive drug, and no significant difference was shown. 

The cough potential period was 24.6 ± 5.6, 40.3 ± 5.4, 32.6 ± 5.1, and 45.3 ± 6.1 s for the blank model, the positive drug group, the NRG, and the NRG NPs, respectively (Figure 4b). The NRG NPs extended the cough potential period by 1.8 times compared with the blank model group (45.3 vs. 24.6, ** *p* < 0.01). Compared with NRG, the cough potential period was 1.4 times higher in the NRG NPs group (45.3 vs. 32.6, ^#^
*p* < 0.05). These results indicated that the NRG NPs had a good cough-relieving effect.

### 2.6. Evaluation of Enzyme Biomarkers in Serum

The PIC is always accompanied by inflammation and oxidative stress, and the NRG exhibits outstanding anti-inflammatory and antioxidant properties [5,51]. Hence, interleukin-6 (IL-6) and C-reactive protein (CRP) as anti-inflammatory enzyme biomarkers, and malondialdehyde (MDA) and superoxide dismutase (SOD) as the antioxidant enzyme biomarkers, were analyzed.

In this in vivo model, levels of anti-inflammatory biomarkers CRP and IL-6 decreased in the positive drug, the NRG, and the NRG NPs groups. The IL-6 concentration was 67.4 ± 10.5, 36.4 ± 4.9, 47.1 ± 9.2, and 28.9 ± 3.8 pg/mL for the blank model, the positive drug group, the NRG, and the NRG NPs, respectively (Figure 5a). Compared with the blank model group, a significant reduction in the IL-6 biomarker concentration in the NRG NPs group was shown (*** *p* < 0.001), indicating that the NRG NPs have good regulatory ability. The difference in the IL-6 concentration with the NRG group was also significant (^#^
*p* < 0.05). However, no significant difference was shown between the NRG NPs and the positive drug groups. A similar tendency was observed for the CRP biomarker; the blank model group showed the highest CRP level (29.0 pg/mL), while the CRP levels of the positive drug, the NRG, and the NRG NPs groups were 23.0, 25.3, and 22.7 pg/mL, respectively. The NRG NPs group presented a significant difference with the blank model group (22.7 vs. 29.0 pg/mL, * *p* < 0.05), and no difference with the positive drug group (22.7 vs. 23.0 pg/mL) (Figure 5b). This result suggests that the inflammatory biomarkers could be regulated effectively by the NRG NPs, indicating that the anti-inflammatory activity of the NRG NPs was higher than that of the free NRG.

The antioxidant activity of the NRG NPs was further studied using the MDA (Figure 5c) and SOD levels (Figure 5d). The blank model group exhibited the highest MDA level (17.4 ± 1.2 nmol/mL). The treatment with the positive drug and the NRG NPs reduced the MDA level to 11.9 ± 0.7 and 12.3 ± 1.3 nmol/mL, respectively. Compared to the blank model group, the NRG NPs significantly decreased the MDA levels (*** *p* < 0.001), but no significant difference was shown compared to the positive drug group. A similar tendency was observed for the SOD levels. The concentrations of SOD were 79.0 ± 1.5, 91.4 ± 2.8, 83.5 ± 1.6, and 90.8 ± 3.5 U/mL for the blank model, positive drug, the NRG, and the NRG NPs groups, respectively. The NRG NPs group exhibited the highest SOD level, which was significantly different from those exhibited by the blank model (** *p* < 0.01), and the NRG (^#^
*p* < 0.05) groups. Thus, the NRG NPs effectively decreased the MDA level and increased the SOD level, and exhibited higher antioxidant activity than the free NRG.

### 2.7. Histological Analysis

The anti-inflammatory effect of the NRG nanoparticles was evaluated via histological analysis of the trachea and lung tissues using hematoxylin-eosin (HE) staining (Figure 6). For the trachea, the blank model group presented lamina propria edema (black arrow) and neutrophil infiltration (yellow arrow), which indicate a significant inflammatory response. Similar results were shown in the positive drug and NRG groups. In contrast, for the NRG NPs group, the epithelial tissue in the tracheal mucosa presented a normal and intact structure, the epithelium had normal morphology and was tightly packed, and no lamina propria edema was shown (Figure 6, first line). These results suggest that the NRG NPs can effectively regulate the inflammatory response. The histological analysis of the lung tissue was further performed. In the model group, many of the lymphocytes (green arrow) and neutrophils (yellow arrow) were observed; these cells were also presented in the positive drug and the NRG groups; additionally, a small amount of blood was observed (black arrow). In the NRG NPs group, only neutrophils were observed. The number of inflammatory cells in the positive drug and the NRG NPs groups was significantly lower than in the model group, suggesting a good anti-inflammatory effect of the NRG NPs. These results are in accordance with a previous report, which reported that the bioactive compounds with antioxidant and anti-inflammatory activities can effectively reduce the lung damage [52]. Based on these phenomena, it can be concluded that the inflammatory response can be regulated effectively by the NRG NPs, further indicating the good anti-inflammatory activity of the NRG NPs.

## 3. Materials and Methods

### 3.1. Materials

Naringenin (NRG, purity > 98%) was purchased from the Aladdin Bio-Chem Technology Co., Ltd. (Shanghai, China), and D-α-tocopherol polyethylene glycol succinate (TPGS; batch number: 20121203) was purchased from Xi′an Healthful Biotechnology Co., Ltd. (Xi’an, China). Pepsin and pancreatin were purchased from Shanghai Macklin Biochemical Co., Ltd. (Shanghai, China). The KH_2_PO_4_ was purchased from Sinopharm Chemical Reagent Co., Ltd. (Beijing, China). The montelukast sodium tablets were purchased from Merck Sharp and Dohme Ltd. (Cramlington, Northumberland, UK). The lipopolysaccharide (LPS) and capsaicin were obtained from the Beijing Solarbio Science & Technology Co., Ltd. (Beijing, China), and thyroxine was purchased from Shenzhen Wenle Biological Technology Co., Ltd. (Shenzhen, China). The IL-6, CRP, MDA, and SOD enzyme-linked immunosorbent assay (ELISA) kits were obtained from Shanghai Enzyme-linked Biotechnology Co., Ltd. (Shanghai, China). All reagents and solvents were used without further purification.

### 3.2. Preparation of NRG Nanoparticles (NPs)

The wet media milling method was utilized to prepare the NRG NPs, and the preparation conditions were optimized as follows: TPGS (10 mg) and NRG (40, 30, 20, or 10 mg) were ultrasonically dispersed in 4 mL water in a vial at 80 °C. After the addition of zirconia beads (0.4–0.6 mm, 5, 8, 10, or 15 g), the solution of the NRG NPs was obtained by stirring at different temperatures (0, 25, 50, 60 °C) and times (1, 2, 3, 4 h). After the process was completed, and the zirconia beads settled, the supernatant was collected with a pipette.

### 3.3. Drug-Loading Content (DLC)

The NRG concentration in the NRG NPs was determined using high-performance liquid chromatography (HPLC; UltiMate 3000; DIONEX) at 25 °C. The samples were separated using a Waters Symmetry C18 column (250 mm × 4.60 mm, 5 μm) at a UV wavelength of 388 nm, a mixed solution containing chromatographic methanol and water (*v*/*v*, 70/30) was used as the eluent, the flow rate was designed as 0.8 mL/min, and the injection volume of the sample solution was 20 μL. After freeze-drying and accurate weighing, the NRG NPs powder (*W*) was dissolved in methanol (*V*). The concentration (*C*) of the NRG was determined by HPLC, and the DLC was calculated as follows:DLC (%) = *V* × *C*/*W* × 100%

### 3.4. Particle Diameter and Morphology

The hydrodynamic diameter and corresponding PDI were determined using dynamic light scattering (DLS, Zetasizer Nano ZS; Malvern Instruments, Malvern, UK) at 25 °C. The concentration of the NRG NPs was approximately 2 mg/mL, and all of the samples were measured three times in parallel.

The NRG NPs (10 μg/mL) were dropped on a 300-mesh copper sheet, and a 2% (*w*/*v*) uranyl acetate solution was used to negatively stain the copper sheet for 30 s. After air drying, the samples were measured using a transmission electron microscope (JEOL Ltd., Tokyo, Japan), the voltage was 120 kV.

### 3.5. Differential Scanning Calorimetry (DSC) Measurements

DSC Q200 (TA Co., New Castle, DE, USA) was applied to perform the DSC measurements. The samples were put into aluminum pans, which were then sealed. Subsequently, the samples were heated and measured under dynamic nitrogen atmosphere in the temperature range of 20–280 °C at a scan rate of 10 °C/min.

### 3.6. X-ray Diffraction (XRD) Measurements

The XRD patterns of the NRG NPs lyophilized powder were conducted using an X-ray diffractometer (D8 Advance, Bruker, Germany). The measurements were carried out with Cu-Kα radiation at a voltage of 40 kV, electric current of 100 mA, and an angular range of 3–80° with a step size of 0.01°.

### 3.7. In Vitro Release Profile

The NRG powder was dispersed in deionized water containing 0.5% CMC-Na to form a suspension. The NRG NPs (1.0 mL) and the NRG powder suspension (1.0 mL) were placed in dialysis bags (MWCO: 8000–14,000, Sigma-Aldrich); then the dialysis bags were put in release media at 37 °C: artificial gastric juice (initial 4 h), artificial intestinal fluid (4–8 h), and PBS solution (8–96 h). The release medium (1.0 mL) was sampled at set time points (0.25, 0.5, 0.75, 1, 2, 4, 6, 8, 10, 12, 24, 36, 48, 72, and 96 h), and the extracted volume was replenished with fresh medium. The sampled media were filtered through 0.22 μm membrane and analyzed via HPLC-UV. The cumulative release was calculated as the ratio of the weight of the released NRG to the total NRG. All experiments were performed in triplicate.

Artificial gastric juice (pH 1.2) was prepared by mixing HCl (9.5%, 16.4 mL) and pepsin (10 g) in distilled water (1 L). Artificial intestinal fluid (pH 6.8) was prepared by adding KH_2_PO_4_ (6.8 g), NaOH (0.1 mol/L), and pancreatin (10 g) to distilled water (1 L).

### 3.8. Animal Experiments

Sixty ICR mice (18–22 g, 6 to 8 weeks old) were purchased from the Beijing Vital River Laboratory Animal Technology Co., Ltd. (Beijing, China). All animals were raised with food and water ad libitum at room temperature in an SPF grade laboratory for one week, and then the post-infectious cough (PIC) model was constructed. Animal experiments were conducted according to the ethical and regulatory guidelines approved by the Animal Ethics Committee of the Peking Union Medical College (Beijing, China). The ethical approval number of this study is SLXD-20201125009.

The PIC mouse model was established according to previously reported studies [53,54]. Briefly, after five days of acclimatization, the mice were exposed to a smoke chamber with four cigarettes for 30 min, which was repeated twice a day for 10 days. On days 11, 14, and 17, the mice were anesthetized and treated with LPS PBS solution (0.4 mg/mL) via nasal drip at a dose of 0.4 µg/g. On days 11–19, the mice were treated with a thyroxine suspension (30 mg/mL, 0.3 mL) via intragastric administration. On days 12, 13, 15, 16, 18, and 19, the mice were treated with capsaicin solution (10^−4^ mol/L, 30.5 mg capsaicin in 1 mL 10% tween-80 solution, 1 mL ethanol, and 8 mL normal saline), the schematic of the entire protocol is shown in Figure 7. The number of coughs exceeded 10 within 3 min, indicating that the PIC model was constructed successfully. The mice were divided into four treatment groups randomly (*n* = 10) and treated as follows: normal saline (model control); montelukast sodium (positive control, 1.2 mg/kg); the NRG (30 mg/kg); and the NRG NPs (30 mg/kg, NRG equivalent concentration). Each group received saline or drugs daily for seven days via intragastric administration. On day 28, the frequency and latency of cough were recorded. Finally, the mice were sacrificed by cervical dislocation, and the blood, trachea, and lung tissues were collected. ELISA detection was implemented totally under the manufacturer’s instructions to determine the concentrations of the relative enzyme biomarkers. The lung and trachea tissues were perfused with 10% formalin, sectioned, and stained with hematoxylin-eosin (HE) dye. 

### 3.9. Statistical Analysis

An independent-samples *t*-test method was used for the statistical analysis of the experimental data (IBM SPSS Statistics software, Version 21, Armonk, New York, NY, USA). Statistical significance was set at * *p* < 0.05.

## 4. Conclusions

In this study, the preparation conditions, including the stirring time, temperature, stirring speed, and media amount, were optimized to obtain the drug-loaded nanoparticles (NPs) with a small diameter and high drug-loading content. Briefly, the naringenin (NRG, 30 mg) and d-α-tocopherol polyethylene glycol succinate (10 mg) were dispersed in deionized water (2 mL). The obtained solution was transferred to a vial containing 10 g zirconia beads and stirred at 50 °C for 2 h at a stirring speed of 300 rpm. The prepared NRG NPs presented a sheet-like morphology with a diameter of approximately 182.2 nm. The in vitro experiments simulating the in vivo conditions showed that the NRG NPs are stable in the gastrointestinal system and are released effectively after entering the blood circulation. The in vivo experiments indicated that the NRG NPs presented good antitussive effects. Meanwhile, the potential period was prolonged 1.8-fold. The anti-inflammatory enzyme biomarkers and histological analysis both revealed that the NRG NPs can regulate the inflammatory and antioxidant stress response effectively. The above results confirmed that the NRG NPs exhibit good oral bioavailability, antitussive effects, anti-inflammatory, and antioxidant effects.

## Figures and Tables

**Figure 1 molecules-27-03736-f001:**
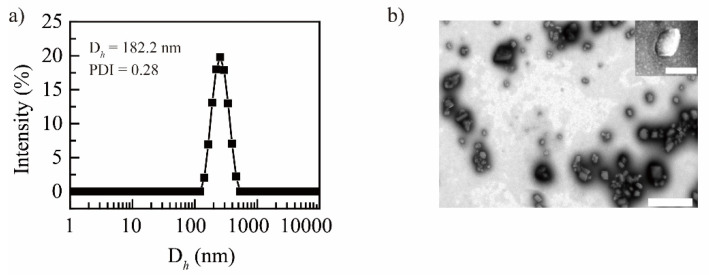
Particle size distribution curve (**a**) and TEM image (**b**) of NRG NPs, scale bar: 500 nm (insert, scale bar: 100 nm).

**Figure 2 molecules-27-03736-f002:**
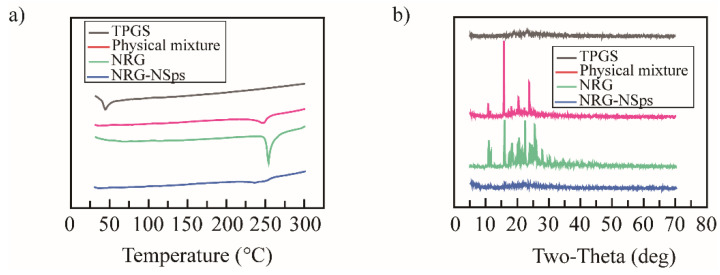
DSC curves (**a**) and XRD curves (**b**) of NRG, TPGS, their physical mixture, and NRG NPs.

**Figure 3 molecules-27-03736-f003:**
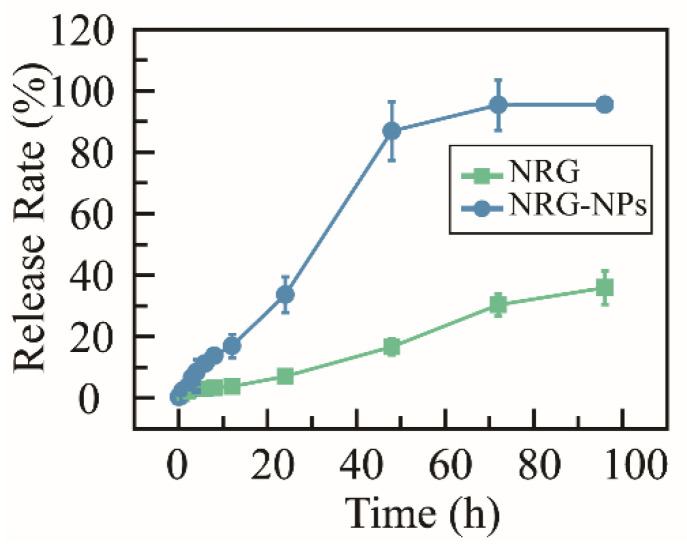
Time-dependent cumulative release profile of NRG NPs and NRG powder at 37 °C, *n* = 3.

**Figure 4 molecules-27-03736-f004:**
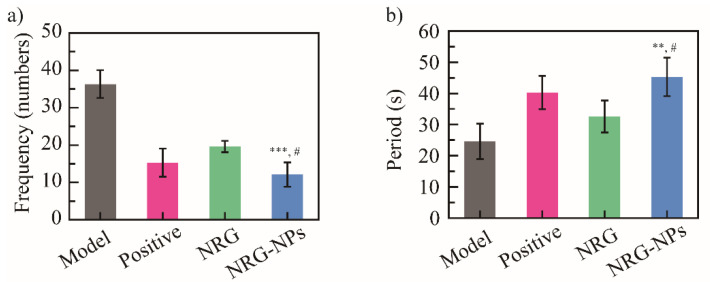
Cough frequency in 5 min (**a**) and cough potential period (**b**) of NRG NPs on PIC mice model (*n* = 10). *** *p* < 0.001, ** *p* < 0.01 vs. blank model group; ^#^ *p* < 0.05 vs. NRG group.

**Figure 5 molecules-27-03736-f005:**
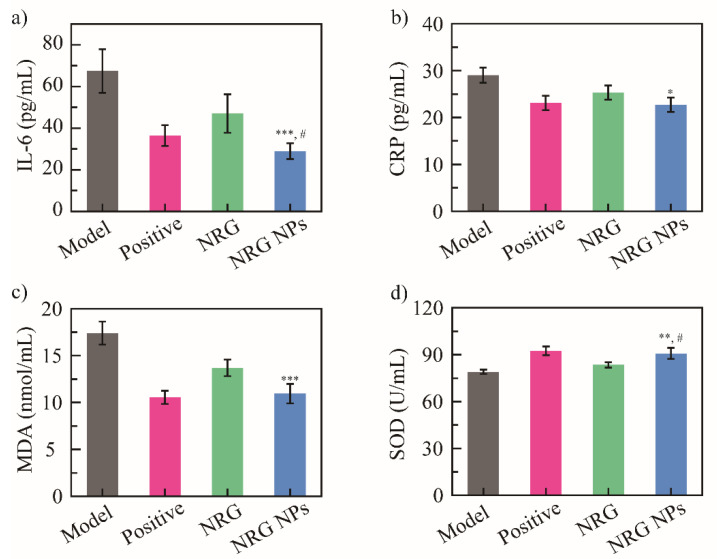
Anti-inflammatory activity: IL-6 biomarker level (**a**), and CRP biomarker level (**b**); antioxidant activity of NRG NPs: MDA biomarker level (**c**), and SOD biomarker level (**d**) (*n* = 5). *** *p* < 0.001, ** *p* < 0.01, * *p* < 0.05 vs. blank model group, ^#^
*p* < 0.05 vs. NRG group.

**Figure 6 molecules-27-03736-f006:**
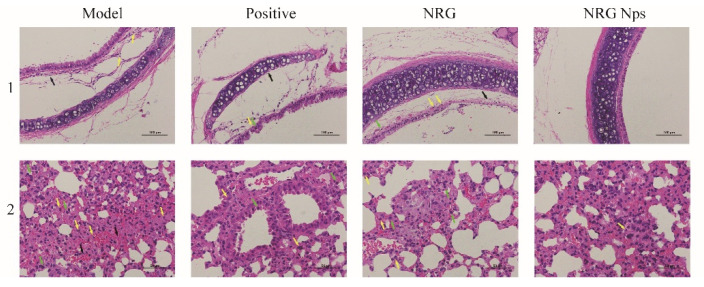
HE staining images of blank model, positive drug, NRG, and NRG NPs groups: trachea tissue (line 1), lung tissue images (line 2).

**Figure 7 molecules-27-03736-f007:**
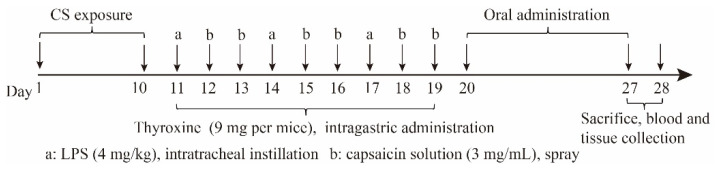
Schematic protocol for establishing the PIC model and the treatment schedule.

**Table 1 molecules-27-03736-t001:** Results of NRG NPs at different FWRs.

Sample	FWR ^a^	D*_h_* ^b^ (nm)	PDI ^c^	DLC ^d^ (%)
1	1/1	214.2 ± 3.3	0.21 ± 0.02	42.3 ± 1.9
2	2/1	227.6 ± 3.2	0.23 ± 0.02	60.0 ± 0.3 ***
3	3/1	225.3 ± 5.6	0.28 ± 0.03	66.1 ± 0.8 ^##^
4	4/1	218.6 ± 5.3	0.32 ± 0.09	64.5 ± 2.9

^a^ Feed weight ratio of drug to nanocarriers; ^b^ hydrodynamic diameter, determined using dynamic light scattering (DLS); ^c^ polydispersity index, detected by DLS; ^d^ drug-loading content; determined using high-performance liquid chromatography (HPLC). (*n* = 3), *** *p* < 0.001, vs. sample 1, ^##^ *p* < 0.01, vs. sample 2.

**Table 2 molecules-27-03736-t002:** Effect of time, temperature, speed, and the amount of milling media on the size and size distribution of NRG NPs.

Sample	Time ^a^ (h)	T ^b^ (°C)	Speed ^c^ (rpm)	W_bead_ ^d^ (g)	D*_h_* (nm)	PDI
1	1	25	200	8	237.1 ± 1.8	0.29 ± 0.06
2	2	25	200	8	229.3 ± 1.8	0.24 ± 0.04
3	3	25	200	8	264.1 ± 1.2	0.26 ± 0.05
4	4	25	200	8	296.6 ± 1.2	0.26 ± 0.02
5	2	0	200	8	267.1 ± 6.2	0.42 ± 0.05
6	2	25	200	8	233.6 ± 4.2	0.27 ± 0.07
7	2	50	200	8	219.9 ± 3.9	0.22 ± 0.04
8	2	60	200	8	285.3 ± 3.1	0.30 ± 0.04
9	2	50	100	8	270.9 ± 2.5	0.27 ± 0.08
10	2	50	200	8	224.9 ± 2.1	0.28 ± 0.04
11	2	50	300	8	209.4 ± 1.5	0.22 ± 0.05
12	2	50	400	8	212.7 ± 1.3	0.21 ± 0.06
13	2	50	300	5	354.2 ± 3.2	0.42 ± 0.02
14	2	50	300	8	208.6 ± 2.1	0.29 ± 0.02
15	2	50	300	10	182.2 ± 3.4	0.28 ± 0.06
16	2	50	300	15	233.5 ± 2.1	0.21 ± 0.05

^a^ Stirring time; ^b^ stirring temperature; ^c^ stirring speed; ^d^ weight of zirconia beads; *n* = 3.

## Data Availability

Not applicable.

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
