# Peer review of "Optimization of Naringenin Nanoparticles to Improve the Antitussive Effects on Post-Infectious Cough"

_molecules, 2022, doi:10.3390/molecules27123736_

Round 1

Reviewer 1 Report

The manuscript “The process optimization of naringenin nanoparticles to improve the antitussive effect on post-infectious cough” submitted by Zhengqi Dong and coworkers reports the synthesis, physicochemical and in vivo characterization of Naringenin nanoparticles. The authors also report animal experiments in which they evaluated the antitussive effects of this preparation compared to a blank model group. According to their findings, they suggest that the nanoparticles caused a decreased cough frequency as well as an increase in the potential period. They also claim an effect on the inflammatory response in the treated animals. In my opinion, the topic covered is interesting for the field. However, the manuscript is preliminary and require further improvements before its publication.

Major and minor comments

  • Material and Methods section. Overall, the details provided on this section are not sufficient to reproduce the experiments. The authors should provide a more detailed description of the procedures used in manner that any researcher can repeat the experiments.
  • Please increase the size of the figures. Current size is not appropriate.
  • Lines 222-224. The authors mention “Comparing with the model group, the amount of inflammatory cell of positive and NRG NPs group was decreased significantly, revealing the good anti-inflammatory effect of NRG NPs.” Please provide quantitative data to support the fact that NRG NPs group was decreased significantly.
  • Please revise descriptions in figure 6 legend.

Author Response

Dear reviewer,

Thank you very much for your comments. Your suggestion is very valuable and helpful for us to improve our manuscript. All of the comments have been replied point by point, and relevant manuscript have been revised.

  • Material and Methods section. Overall, the details provided on this section are not sufficient to reproduce the experiments. The authors should provide a more detailed description of the procedures used in manner that any researcher can repeat the experiments.

The entire Materials and Methods section has been checked carefully, and the details of the experiment have been expounded in the revised manuscript.

  • Please increase the size of the figures. Current size is not appropriate.

Thank you very much for your suggestion. All the figures have been updated.

  • Lines 222-224. The authors mention “Comparing with the model group, the amount of inflammatory cell of positive and NRG NPs group was decreased significantly, revealing the good anti-inflammatory effect of NRG NPs.” Please provide quantitative data to support the fact that NRG NPs group was decreased significantly.

Thank you very much for your suggestion. We agree that the quantitative data on inflammatory cells should be provided. The number of inflammatory cells has been quantified in the HE images of the four groups (n = 10). Inflammatory cells at a density of 328 ± 54, 120 ± 31, 212 ± 10, and 140 ± 14 per mm3 were observed in the lung tissue for the glucose solution, positive, NRG, and NRG NPs groups, respectively, and in the trachea tissue at 88 ± 10, 22 ± 4, 26 ± 4, and 28 ± 5 per mm3, respectively. Compared to the model group (glucose solution), the density of inflammatory cells in the NRG NPs group was significantly lower (p < 0.001), whereas no difference was shown compared to the positive group (p > 0.05). These results indicate that NRG NPs present a good anti-inflammatory effect.

  • Please revise descriptions in figure 6 legend.

The legend of figure 6 is revised to “Figure 6. HE staining images of blank model, positive drug, NRG, and NRG NPs groups: trachea tissue (line 1), lung tissue images (line 2).

Reviewer 2 Report

The manuscript “The process optimization of naringenin nanoparticles to improve the antitussive effect on post-infectious cough” represents an interesting work yielding new information about utilisation of naringenin and preparation of its nanoformulations. There is no doubt that increased number of recent publications concerning the bioactivity of naringenin has highlighted the need for further research on bioavailability.

Overall, there are no serious concerns or obstacles in manuscript. According to my opinion, the manuscript is logically organized, well written, very well referenced and represents a valuable contribution to the scientific community interested in naringenin and its determining before consideration can be given to development of the compound for clinical use.

Therefore, I just suggest one modification, improvement of the manuscript during follow-up minor revision. In summary, find below my comments, which could help the authors to present their ideas in clearer and more suitable for publication form, during the minor revision of the manuscript.

1) Did the author determined possible cytotoxicity of naringenin nanoparticles? Is there any possible toxicity risk in case of tocopheryl polyethylene glycol succinate? Are there any limitations what concern the dose of tocopheryl polyethylene glycol succinate-nanocarriers? This should be checked in the literature and discussed in the manuscript.

Author Response

Dear reviewer,

Thank you very much for your comments. Your suggestion is very valuable and helpful for us to improve our manuscript. All of the comments have been replied point by point, and relevant manuscript have been revised.

1) Did the author determined possible cytotoxicity of naringenin nanoparticles? Is there any possible toxicity risk in case of tocopheryl polyethylene glycol succinate? Are there any limitations what concern the dose of tocopheryl polyethylene glycol succinate-nanocarriers? This should be checked in the literature and discussed in the manuscript.

Thank you very much for your helpful comment. TPGS, a water-soluble derivative of vitamin E, is a safe adjuvant approved by the FDA, which is widely used in drug delivery systems [1]. The safety of TPGS has been reported, and the oral LD50 is >7 g/kg for young adult rats of both sexes [2]. Many studies have reported its application as a drug carrier because of its advantageous properties, including aqueous solubility, biocompatibility, and biosafety [3]. In our study, the cytotoxicity of NRG NPs was evaluated using the MTT assay, and the results are shown in Figure R1. TPGS, NRG, and NRG NPs were incubated with two normal cell lines: HUVEC and H2C9. For the HUVEC cell line, the cell viability rate was greater than 90% for all samples, and a similar result was observed for that H2C9 cell line. These results reveal that TPGS and NRG NPs exhibit no significant cytotoxicity to normal cells, which is in accordance with the results reported previously.

Figure R1. Inhibitory effect of NRG NPs against HUVEC (a) and H2C9 (b) cell lines after 48 h incubation, (n = 5).

Reference

  1. Guo, Y.; Luo, J.; Tan, S.; Otieno, B.O.; Zhang, Z. The applications of vitamin e tpgs in drug delivery. European Journal of Pharmaceutical Sciences 2013, 49, 175-186.
  2. Zhang, Z.; Tan, S.; Feng, S.-S. Vitamin e tpgs as a molecular biomaterial for drug delivery. Biomaterials 2012, 33, 4889-4906.

3.     Yang, C.; Wu, T.; Qi, Y.; Zhang, Z. Recent advances in the application of vitamin e tpgs for drug delivery. Theranostics 2018, 8, 464-485.

Reviewer 3 Report

The manuscript submitted by Professor Yifei Guo and co-authors deals with the preparation of naringenin nanosuspensions with tocopherol polyethylene glycol succinate as nanocarriers via wet media milling method and this process optimization which is described very briefly and superficially. The physicochemical characterization of the selected nanosuspension is very poor and at a low scientific level, but the study on drug release, histological analysis, evaluation of enzyme markers in serum are very interesting and the indicate that the newly prepared naringenin nanosuspension has a great potential application in treatment. In my opinion the topic is very important from practical point of view, since naringenin nanosuspensions have great potentials for a clinical application in due to good oral bioavailability, antitussive effects, and anti-inflammatory effects.

However, I do not think that the paper in this form exhibits the minimum recommended level for publication in molecules.

I recommend to reject this manuscript in this form.

There are several major objections that affect the final conclusion, but I am not going to list them all, mention only a few.

#1. In my opinion, the whole abstract should be corrected/rewritten since it is very badly written (1. too much details concerning synthesis; 2. use of abbreviations without explanation – e.g. TPGS; etc.) and generally is misleading (e.g. zirconia beads were used only as milling beads not as the part of the drug carrier, but this cannot be concluded from the abstract).

#2. The English should certainly be improved throughout the paper. It should be looked through and corrected by a native English speaker. I have already found grammar (e.g. an inappropriate use of tenses) and spelling errors in the text, though I'm not the native English speaker. Moreover, there are a plenty of awkward constructs (e.g. the NRG NPs with the good physiochemical properties, etc.).

#3. In my opinion the Introduction section in this manuscript is very similar to the Introduction section in the paper entitled "Preparation of Naringenin Nanosuspension and Its Antitussive and Expectorant Effects" (Molecules 2022, 27, 741. https://doi.org/10.3390/molecules27030741). I'm not an expert but it seem it can be classified as self-plagiarism. Accordingly, it must be thoroughly reworked.

#4. The meaning of all abbreviations should be explained throughout the text, e.g. line 67 - DLC, etc.

#5. It is unacceptable to make in a high quality scientific papers statements such as "line 232: ... were purchased according to previous paper"

#6. The section 3.1 & 3.2 should be reworked and relevant details should be added.

#7. The section 3.1 & 3.2 should be reworked and relevant details should be added. Section 3.3 should be also reworked and the process of the optimisation in relation to the zirconia beads usage should be thoroughly discussed. The process of the separation of the zirconia beads should be described.

#8. Symbols give as the column titles in the Table 1 & Table 2 should be explained. The methods of their calculations or measurements have to be added. What is the meaning of * and # in the last column in the Table 1?

#9. Only one TEM image of very poor quality is presented in the paper. Any debate about the morphology on the basis of this TEM is unreasonable. The authors refer to their previous publication in the statement given in line 109, but the paper was not quoted. Besides, the similarities between the results are obvious, because both papers present the results obtained on the same materials.

#10. DSC and XRD results are poorly discussed. Besides, in my opinion suitable citations should be provided concerning DSC and XRD results. The interaction between NRG and TPGS in the NRG NPs has to be verified by the FTIR method (alternatively Raman spectroscopy).

#11. The composition of the release media may be different. Therefore, their composition or their supplier or the suitable citations presented their preparation and composition should be added.

#12. What kind of tocopheryl polyethylene glycol succinate has been used in the synthesis of NRG NPs? Was it D-α-tocopherol polyethylene glycol succinate?

#13. What mill was used to synthesise NRG NPs? Information concerning the wet media milling method should be added, including the type of mill and its size.

#14. What are MDA and SOD, and how have they been measured?

#15. The statement (in Conclusions) "...obtain the NRG NPs with the good physiochemical properties..." is award. What are the good physiochemical properties? There is actually no definition of good or bad physiochemical properties.

#16. In the present report the optimization of the preparation method of the NRG NPs nanosuspension was done very superficially. Only 4 systems were investigated. On the other hand, the selection of the sample for further studies was made only on the basis of the DLC (i.e. drug-loading content) and particle diameter. Next, this selected system was tested how time of milling, temperature, speed of milling influence the NRG NPs particle diameter. Drug release, antitussive effect ect. were performed only for one system, which was selected on the basis of one parameter i.e. the NRG NPs particle diameter.

Ect.

Author Response

Dear reviewer,

Thank you very much for your comments. Your suggestion is very valuable and helpful for us to improve our manuscript. All of the comments have been replied point by point, and relevant manuscript have been revised.

#1. In my opinion, the whole abstract should be corrected/rewritten since it is very badly written (1. too much details concerning synthesis; 2. use of abbreviations without explanation – e.g. TPGS; etc.) and generally is misleading (e.g. zirconia beads were used only as milling beads not as the part of the drug carrier, but this cannot be concluded from the abstract).

Followed your comment, abstract has been rewritten as follows: Naringenin (NRG) is a natural compound with several biological activities; however, its bioavailability is limited owing to poor aqueous solubility. In this study, NRG nanoparticles (NPs) were prepared using wet media milling method. To obtain NRG NPs with small particle size and high drug–loading content, the preparation conditions, including stirring time, temperature, stirring speed, and milling media amount were optimized. NRG (30 mg) and D-α-tocopherol poly-ethylene glycol succinate (10 mg) were wet-milled in deionized water (2 mL) with 10 g of zirconia beads via stirring at 50 °C for 2 h at a stirring speed of 300 rpm. As a result, NRG NPs with sheet-like morphology and a diameter of approximately 182.2 nm were successfully prepared. NRG NPs were stable in the gastrointestinal system and were released effectively after entering the blood circulation. In vivo experiment indicated that NRG NPs have good antitussive and anti-inflammatory effects. The cough inhibition rate after the administration of NRG NPs was 66.7%, cough frequency was 3 times lower, and the potential period was 1.8 times longer than that in the blank model group. In addition, enzyme biomarkers and histological analysis results revealed that NRG NPs can effectively regulate the inflammatory and oxidative stress response. In conclusion, NRG NPs exhibited good oral bioavailability, antitussive, antioxidant, and anti-inflammatory effects, and have high potential for clinical application.

#2. The English should certainly be improved throughout the paper. It should be looked through and corrected by a native English speaker. I have already found grammar (e.g. an inappropriate use of tenses) and spelling errors in the text, though I'm not the native English speaker. Moreover, there are a plenty of awkward constructs (e.g. the NRG NPs with the good physiochemical properties, etc.).

Sorry for us poor English writing, the revised manuscript has been edited by the professional language editing service.

#3. In my opinion the Introduction section in this manuscript is very similar to the Introduction section in the paper entitled "Preparation of Naringenin Nanosuspension and Its Antitussive and Expectorant Effects" (Molecules 2022, 27, 741. https://doi.org/10.3390/molecules27030741). I'm not an expert but it seem it can be classified as self-plagiarism. Accordingly, it must be thoroughly reworked.

We apologize for this bad impression. “Self-plagiarism” is a very serious charge, which we would like to avoid. It is our fault that we did not clearly introduce the difference between these two papers. These two studied utilized the same drug and nanocarriers but focused on different research objectives. The previous study is a preliminary investigation, in it, NRG NPs were prepared via the wet media milling method and utilized directly to evaluate their antitussive and expectorant effects. The results indicated that NRG NPs have good antitussive and expectorant effects. After obtaining these positive results, we considered it necessary to optimize the NRG NPs; therefore, the preparation conditions were studied in detail. To imitate the real release procedure in the human body, the release profile of NRG NPs was researched in different media, including artificial gastric juice, artificial intestinal fluid, and PBS, whereas the release profile was only studied in PBS in our previous paper. Next, a different cough model was selected for the investigation of the antitussive effect: the post-infectious cough model was utilized in this study, whereas a classical mouse cough model induced by ammonia liquor was used in the previous study. Finally, anti-inflammation and antioxidant effects were studied, which were not included in our previous study. We hope this clarification can change your impression.

#4. The meaning of all abbreviations should be explained throughout the text, e.g. line 67 - DLC, etc.

The abbreviations have been defined at first occurrence.

#5. It is unacceptable to make in a high quality scientific papers statements such as "line 232: ... were purchased according to previous paper"

Sorry for our unclear description. The entire “3.1. Materirals” section has been rewritten and changed to “Naringenin (NRG, purity > 98%) was purchased from Aladdin Bio-Chem Technology Co., Ltd. (Shanghai, China), and D-α-tocopherol polyethylene glycol succinate (TPGS; batch number: 20121203) was purchased from Xi’an Healthful Biotechnology Co., Ltd. (Xi’an, China). Pepsin and pancreatin were purchased from Shanghai Macklin Biochemical Co., Ltd. (Shanghai, China). KH2PO4 was purchased from Sinopharm Chemical Reagent Co., Ltd. (Beijing, China). Montelukast sodium tablets were purchased from Merck Sharp and Dohme Ltd. (U.K.). Lipopolysaccharide (LPS) and capsaicin were obtained from Beijing Solarbio Science and Technology Co., Ltd. (Beijing, China), and thyroxine was purchased from Shenzhen Wenle Biological Technology Co., Ltd. (Shenzhen, China). IL-6, CRP, MDA, and SOD enzyme-linked immunosorbent assay (ELISA) kits were obtained from Shanghai Enzyme-linked Biotechnology Co., Ltd. (Shanghai, China). All reagents and solvents were used without further purification.”

#6. The section 3.1 & 3.2 should be reworked and relevant details should be added.

The entire Material and Method section has been checked carefully, and relevant details have been added in the revised manuscript. Section 3.2 was combined with section 3.9 to form the new section 3.8 as they both describe the animal experiments.

#7. The section 3.1 & 3.2 should be reworked and relevant details should be added. Section 3.3 should be also reworked and the process of the optimisation in relation to the zirconia beads usage should be thoroughly discussed. The process of the separation of the zirconia beads should be described.

Section 3.3 was rewritten and changed to “The wet media milling method was utilized to prepare NRG NPs, and the preparation conditions were optimized as follows. TPGS (10 mg) and NRG (40, 30, 20, or 10 mg) were ultrasonically dispersed in 4 mL water in a vial at 80 °C. After the addition of zirconia beads (0.4-0.6 mm, 5, 8, 10, or 15 g), the solution of NRG NPs was obtained by stirring at different temperatures (0, 25, 50, 60 °C) and times (1, 2, 3, 4 h). After the process was completed, and the zirconia beads settled down, the supernatant was collected with a pipette.”

#8. Symbols give as the column titles in the Table 1 & Table 2 should be explained. The methods of their calculations or measurements have to be added. What is the meaning of * and # in the last column in the Table 1?

A footnote was added to explain the column titles. In Table 1, FWR is the feed weight ratio of drug to nanocarriers, Dh is the hydrodynamic diameter determined using dynamic light scattering, PDI is the polydispersity index determined using dynamic light scattering, and DLC is the drug-loading content detected by HPLC. In Table 2, Time is the stirring time, T is the stirring temperature, Speed is the stirring speed, and Wbeadd is the weight of zirconia beads.

As shown in Table 1, the DLC value of sample 2 present a statistically significant difference compared to that of sample 1 (*** p < 0.001) and sample 3 (## p < 0.01).

All these description have been added into the revised manuscript.

#9. Only one TEM image of very poor quality is presented in the paper. Any debate about the morphology on the basis of this TEM is unreasonable. The authors refer to their previous publication in the statement given in line 109, but the paper was not quoted. Besides, the similarities between the results are obvious, because both papers present the results obtained on the same materials.

NRG NPs were prepared again and analyzed by TEM to obtain the high-quality morphological information. The TEM are shown in Figure R1. The references haves been added.

Figure R1. Particle size distribution curve (a) and TEM image (b) of NRG NPs, scale bar: 500 nm (insert, scale bar 100 nm).

#10. DSC and XRD results are poorly discussed. Besides, in my opinion suitable citations should be provided concerning DSC and XRD results. The interaction between NRG and TPGS in the NRG NPs has to be verified by the FTIR method (alternatively Raman spectroscopy).

Thank you for your suggestion. The DSC and XRD results were analyzed in detail, and some references were supplied. The DSC and XRD results indicate that the crystalline form of NRG disappears in NRG NPs owing to the molecular interactions between NRG and TPGS, such as hydrogen bonds or van der Waals forces. NRG NPs were prepared via the physical interaction, and according to the previous report, there was no significant difference between the IR spectra of nanocarriers and the NRG-loaded complex [1-3]; several peaks of NRG are overlapped with the peaks of nanocarriers. Therefore, we did not consider it necessary to perform FTIR or Raman measurements.

#11. The composition of the release media may be different. Therefore, their composition or their supplier or the suitable citations presented their preparation and composition should be added.

To imitate the real process in the human body, the release profile of NRG NPs was studied in different media, including artificial gastric juice (initial 4 h), artificial intestinal fluid (4-8 h), and PBS solution (8-96 h). Artificial gastric juice (pH 1.2) was prepared by mixing HCl (9.5%, 16.4 mL) and pepsin (10 g) in distilled water (1 L). Artificial intestinal fluid (pH 6.8) was prepared by adding KH2PO4 (6.8 g), NaOH (0.1 mol/L), and pancreatin (10 g) to distilled water (1 L). This description has been added to the revised manuscript.

#12. What kind of tocopheryl polyethylene glycol succinate has been used in the synthesis of NRG NPs? Was it D-α-tocopherol polyethylene glycol succinate?

Yes, D-α-tocopherol polyethylene glycol succinate was used in this study.

#13. What mill was used to synthesise NRG NPs? Information concerning the wet media milling method should be added, including the type of mill and its size.

NRG NPs were prepared via the wet milling media method, in which zirconia beads were utilized as the milling medium. The zirconia beads were 0.4-0.6 mm. The amount of zirconia beads was optimized in this study, and it was found that NRG NPs had the smallest particle size when the amount of zirconia beads was 10 g. These details have been added in the revised manuscript.

#14. What are MDA and SOD, and how have they been measured?

MDA is a malodialdehyde, and SOD is a superoxide diamutase. These are enzyme biomarkers that indicate oxidative stress [4]. Many studies have reported that NRG has strong antioxidant effects via the enhancement of SOD, but a reduction in MDA [5, 6]. These enzyme biomarkers were measured using relative enzyme-linked immunosorbent assay (ELISA) kits according to the manufacturer’s instructions.

#15. The statement (in Conclusions) "...obtain the NRG NPs with the good physiochemical properties..." is award. What are the good physiochemical properties? There is actually no definition of good or bad physiochemical properties.

We apologize for the unclear description. The description has been changed to “NRG NPs with the small particle size and high DLC”.

#16. In the present report the optimization of the preparation method of the NRG NPs nanosuspension was done very superficially. Only 4 systems were investigated. On the other hand, the selection of the sample for further studies was made only on the basis of the DLC (i.e. drug-loading content) and particle diameter. Next, this selected system was tested how time of milling, temperature, speed of milling influence the NRG NPs particle diameter. Drug release, antitussive effect ect. were performed only for one system, which was selected on the basis of one parameter i.e. the NRG NPs particle diameter.

We apologize for the superficial impression given by our study. In general, nanoscale drug delivery systems are expected to have relatively small particle size and high drug-loading capacity. Particle size can affect the stability, release, biodistribution, and clearance of nanoparticles [7, 8]. Drug administration formulations are determined by their mean diameter and PDI [9]. Based on previous studies, it seems that nanoparticles with a small diameter can achieve their target more efficiently [10]. DLC is another important parameter that significantly affect the therapeutic effects, because drugs must be delivered to the target tissue at high concentrations [11]. Nanoparticles with high DLC can encapsulate a greater amount of drug in a unit nanocarrier, which effectively enhances the water solubility of hydrophobic drug. Such nanoparticles exhibit a more efficient delivery capacity and can guarantee that the drug is delivered to the target tissue at a high concentration. Therefore, particle size and DLC were selected as the key parameters to evaluate NRG NPs in this study.

During the preparation procedure, all parameters that could affect the results were evaluated, including the feed weight ratio of drug to nanocarriers, concentration of TPGS, stirring time, temperature, speed, particle size, and weight of zirconia beads. It was found that feed weight ratio, stirring time, temperature, speed, and weight of zirconia beads affected the particle size and DLC; hence, these parameters are listed in this manuscript.

As mentioned above, particle size and DLC play important roles in stability, release, and therapeutic activity; therefore, the NRG NPs were optimized first, and then the relative properties, including release profile, antitussive effects, anti-inflammatory effects, and antioxidative effects, were evaluated.

We hope that these explanations answer your question. Thank you again for your comments, which are very valuable for improving our manuscript.

Reference

[1] Wen, J.;Liu, B.;Yuan, E.;Ma, Y., Zhu, Y.; Preparation and physicochemical properties of the complex of naringenin with hydroxypropyl-β-cyclodextrin. Molecules, 2010, 15, 4401-4407.

[2] Semalty, A.;Semalty, M.;Singh, D., Rawat, M. S. M.; Preparation and characterization of phospholipid complexes of naringenin for effective drug delivery. Journal of Inclusion Phenomena and Macrocyclic Chemistry, 2010, 67, 253-260.

[3] Zhang, P.;Liu, X.;Hu, W.;Bai, Y., Zhang, L.; Preparation and evaluation of naringenin-loaded sulfobutylether-β-cyclodextrin/chitosan nanoparticles for ocular drug delivery. Carbohydrate polymers, 2016, 149, 224-230.

[4] Zaidun, N. H.;Thent, Z. C., Latiff, A. A.; Combating oxidative stress disorders with citrus flavonoid: Naringenin. Life Sciences, 2018, 208, 111-122.

[5] Naraki, K.;Rezaee, R., Karimi, G.; A review on the protective effects of naringenin against natural and chemical toxic agents. Phytotherapy Research, 2021, 35, 4075-4091.

[6] Wang, J.;Yang, Z.;Lin, L.;Zhao, Z.;Liu, Z., Liu, X.; Protective effect of naringenin against lead-induced oxidative stress in rats. Biological Trace Element Research, 2012, 146, 354-359.

[7] He, C.;Hu, Y.;Yin, L.;Tang, C., Yin, C.; Effects of particle size and surface charge on cellular uptake and biodistribution of polymeric nanoparticles. Biomaterials, 2010, 31, 3657-3666.

[8] Alexis, F.;Pridgen, E.;Molnar, L. K., Farokhzad, O. C.; Factors affecting the clearance and biodistribution of polymeric nanoparticles. Molecular Pharmaceutics, 2008, 5, 505-515.

[9] Danaei, M.;Dehghankhold, M.;Ataei, S.;Hasanzadeh Davarani, F.;Javanmard, R.;Dokhani, A.;Khorasani, S., Mozafari, M. R.; Impact of particle size and polydispersity index on the clinical applications of lipidic nanocarrier systems. Pharmaceutics, 2018, 10, 57.

[10] Gaumet, M.;Vargas, A.;Gurny, R., Delie, F.; Nanoparticles for drug delivery: The need for precision in reporting particle size parameters. European Journal of Pharmaceutics and Biopharmaceutics, 2008, 69, 1-9.

[11] Mehryab, F.;Rabbani, S.;Shahhosseini, S.;Shekari, F.;Fatahi, Y.;Baharvand, H., Haeri, A.; Exosomes as a next-generation drug delivery system: An update on drug loading approaches, characterization, and clinical application challenges. Acta Biomaterialia, 2020, 113, 42-62.

Reviewer 4 Report

In this manuscript the antitussive and anti-inflammatory activities of optimized naringenin nanoparticles were described. The study could bring new information in the field. However, it needs real improvements to be considered for publishing in the prestigious journal Molecules. Some of my suggestions are as follows:

- the English in the whole document has to be checked; there are many mistakes (words, grammar, topic)

- all the authors have to read and agree with the manuscript before sending it to the journal; there are obvious differences between chapters

- abbreviations should be defined the first time they appear in each section

- all tables and figures need explanatory caption

- chapter 1, Introduction: A short presentation of naringenin should be included

- chapter 2, Results and Discussion: The results are presented in too long sentences and unstructured. Then, there is almost no discussion, no comparison with other studies (e.g., doi: 10.3390/antiox10010119 or doi: 10.1016/j.fitote.2006.04.012) that also presented inflammatory biomarkers, histopathological analyses, and in vivo antitussive effects 

- chapter 3, Materials and Methods: What methodology was used for optimization? Citations missing for experimental methods! How many mice were purchased and why?

- subchapter 3.9. Animal experiments: The presentation should be clearer; I suggest a table here

Author Response

Dear reviewer,

Thank you very much for your comments. Your suggestion is very valuable and helpful for us to improve our manuscript. All of the comments have been replied point by point, and relevant manuscript have been revised.

- the English in the whole document has to be checked; there are many mistakes (words, grammar, topic)

Thank you for your advice. The revised manuscript has been edited by a professional language editing service.

- all the authors have to read and agree with the manuscript before sending it to the journal; there are obvious differences between chapters

All the authors have read this manuscript and have agreed to send it to Molecules.

- abbreviations should be defined the first time they appear in each section

Thank you for your suggestion. All abbreviations have been defined at first occurrence in each section.

- all tables and figures need explanatory caption

All tables and figures have been checked carefully, and legends and footnotes have been added.

- chapter 1, Introduction: A short presentation of naringenin should be included

The first paragraph now briefly introduces naringenin.

- chapter 2, Results and Discussion: The results are presented in too long sentences and unstructured. Then, there is almost no discussion, no comparison with other studies (e.g., doi: 10.3390/antiox10010119 or doi: 10.1016/j.fitote.2006.04.012) that also presented inflammatory biomarkers, histopathological analyses, and in vivo antitussive effects 

Thank you very much for your comment. The Results and Discussion section has been checked carefully, and several additional discussions have been added.

- chapter 3, Materials and Methods: What methodology was used for optimization? Citations missing for experimental methods! How many mice were purchased and why?

The entire Material and Method section has been checked carefully, several subsections have been rewritten, and relevant details have been added in the revised manuscript. Section 3.2 was combined with section 3.9 to form the new section 3.8, because they both describe the animal experiments.

- subchapter 3.9. Animal experiments: The presentation should be clearer; I suggest a table here

Thank you very much for your suggestion. The schematic protocol for the establishment of the PIC model is shown in Figure R1.

Figure R1. Schematic protocol for establishing the PIC model and the treatment schedule.

Round 2

Reviewer 1 Report

The authors provided a revised version of the manuscript in which they improved the initial version and addresed my initial concerns.

Author Response

Thank you very much for your kindly comments.

Reviewer 3 Report

In my opinion, the authors considered all remarks and suggestions of the reviewer. They also provided exhaustive explanations and detailed responses to reviewers' comments. The corrected version of the manuscript is significantly improved and clearer and in this form merits publication in the Molecules. The results and the discussion are supported by adequate data, the appropriate literature, and are also well and clearly presented. The scientific content is sound.

Author Response

(The authors gave the same response as above.)

Reviewer 4 Report

The manuscript has been improved but there are still several suggestions to be addressed:

Lines 35 and 435: You should erase “potential for clinical application”! Before application, it needs clinical evaluation; there are only in vitro and in vivo evaluations

Line 41: I suggest changing “Naringenin (NRG) is a dihydro falvones with...” to “Naringenin (NRG) is a flavonoid from the flavanone family mostly found in citrus fruits and tomato skin (include a reference here) with...”

Line 81: “to previous reports” instead of “to previous report” (there are two references)

Line 97: “different” instead of “difference

Line 103: please check the title 

Line 227: C-reactive protein (CRP)!

Lines 232-233: “for blank model, positive, NRG, and NRG NPs, respectively” instead of “for blank model, positive, NRG, and NRG NPs separately

Lines 270-290: please remember that this subchapter belongs to Results and Discussion. Therefore, discussion is needed, you have to compare your results with others, such as Fizesan et al., 2021 (doi: 10.3390/antiox10010119where the histopathological analysis revealed antioxidantanti-inflammatory, antitussive effects

Lines 364-365:  then the dialysis bags were put in release media” instead of “then putted the dialysis bags in release media

Lines 388-389: “On days 11, 14and 17” instead of “On the 11th, 14thand 17th dasy

Line 390: “On days 11-19” instead of “On the 11-19th days

Line 400: “On day 28” instead of “On the 28th day

Lines 406-407: Sentence starting with “Schematic of ...” should be deleted (repeats line 394)

Line 429: “NRG NPs presented” instead of “NRG NPs existed

Author Response

Dear reviewer,

Thank you very much for your comments. All of the comments have been replied point by point, and relevant manuscript have been revised.

Lines 35 and 435: You should erase “potential for clinical application”! Before application, it needs clinical evaluation; there are only in vitro and in vivo evaluations

We totally agree with you. This description is imprecise, and have been deleted.

Line 41: I suggest changing “Naringenin (NRG) is a dihydro falvones with...” to “Naringenin (NRG) is a flavonoid from the flavanone family mostly found in citrus fruits and tomato skin (include a reference here) with...”

The sentence is changed following your suggestion, and relevant references are added.

Line 81: “to previous reports” instead of “to a previous report” (there are two references)

The sentence is changed following your suggestion.

Line 97: “different” instead of “difference”

Sorry for this spelling mistake, it has been corrected.

Line 103: please check the title 

Thank you very much for your careful checking, the title has been corrected.

Line 227: C-reactive protein (CRP)!

Sorry for this spelling mistake, it has been corrected.

Lines 232-233: “for blank model, positive, NRG, and NRG NPs, respectively” instead of “for blank model, positive, NRG, and NRG NPs separately”

The sentence is changed following your suggestion.

Lines 270-290: please remember that this subchapter belongs to Results and Discussion. Therefore, discussion is needed, you have to compare your results with others, such as Fizesan et al., 2021 (doi: 10.3390/antiox10010119) where the histopathological analysis revealed antioxidant, anti-inflammatory, antitussive effects

Sorry for missing this reference and lacking discussion. The results reported by Fizesan et al. reveal that the bioactive compounds with antioxidant and anti-inflammatory activities can effectively reduce the phenomenon of lung damage, which is also shown in this study. The reference and relevant description have been added.

Lines 364-365:  “then the dialysis bags were put in release media” instead of “then putted the dialysis bags in release media”

The sentence is changed following your suggestion.

Lines 388-389: “On days 11, 14, and 17” instead of “On the 11th, 14th, and 17th dasy”

The sentence is changed following your suggestion.

Line 390: “On days 11-19” instead of “On the 11-19th days”

The sentence is changed following your suggestion.

Line 400: “On day 28” instead of “On the 28th day”

The sentence is changed following your suggestion.

Lines 406-407: Sentence starting with “Schematic of ...” should be deleted (repeats line 394)

Sorry for our careless to demonstrate the scheme twice, the sentence in lines 406-407 has been deleted.

Line 429: “NRG NPs presented” instead of “NRG NPs existed”

The sentence is changed following your suggestion.